# Withanolide Metabolites Inhibit PI3K/AKT and MAPK Pro-Survival Pathways and Induce Apoptosis in Acute Myeloid Leukemia Cells

**DOI:** 10.3390/biomedicines8090333

**Published:** 2020-09-06

**Authors:** Nosheen Akhtar, Muhammad Waleed Baig, Ihsan-ul Haq, Vinothini Rajeeve, Pedro Rodriguez Cutillas

**Affiliations:** 1Cell Signalling and Proteomics Group, Centre of Haemato-Oncology, Barts Cancer Institute, Queen Mary University of London, London EC1M 6BQ, UK; v.rajeeve@qmul.ac.uk; 2Department of Biological Sciences, National University of Medical Sciences, Rawalpindi 46000, Pakistan; 3Department of Pharmacy, Quaid-e-Azam University, Islamabad 45320, Pakistan; mwbg7@yahoo.com (M.W.B.); ihsn99@yahoo.com (I.-u.H.)

**Keywords:** withametelin, coagulansin A, acute myeloid leukemia, proteomics, PI3K pathway, MAPK pathway, apoptosis

## Abstract

Acute myeloid leukemia (AML) is an aggressive disease and, despite advances, its treatment remains challenging. Therefore, it remains important to identify new agents for the management of this disease. Withanolides, a group of steroidal lactones found in Solanaceae plants are of potential interest due to their reported anticancer activities in different settings. In this study we investigated the anti-proliferative effects and mode of action of Solanaceae-derived withanolides in AML cell models; these metabolites include withametelin (WTH) and Coagulansin A (CoA) isolated from *Datura innoxia* and *Withania coagluanse*, respectively. Both withanolides inhibited the proliferation of AML cells and induced cell death, with WTH being more potent than CoA in the AML models tested. Quantitative label-free proteomics and phosphoproteomics were employed to define the mechanism of action of the studied withanolides. We identified and quantified 5269 proteins and 17,482 phosphosites in cells treated with WTH, CoA or vehicle control. Withanolides modulated the expression of proteins involved in regulating key cellular processes including cell cycle, metabolism, signaling, protein degradation and gene expression. Enrichment analysis of the phosphoproteomics data against kinase substrates, kinase-kinase relationships and canonical pathways showed that the withanolides decreased the activity of kinases such as phosphoinositide 3-kinase (PI3K), protein kinase B (PKB; also known as RAC-alpha serine/threonine-protein kinase or AKT), mammalian target of rapamycin (mTOR), extracellular signal-regulated protein kinase 1 and 2 (ERK1/2) and the serine/threonine-protein kinase A-Raf (ARAF), while increasing the activation of DNA repair kinases. These results indicate that withanolide metabolites have pleiotropic effects in the modulation of oncogenic pro-survival and pro-apoptotic signaling pathways that regulate the induction of apoptosis. Withanolide mediated apoptosis was confirmed by immunoblotting showing increased expression of cleaved PARP and Caspases 3, 8 and 9 as a result of treatment. Overall, our results suggest that WTH and CoA have therapeutic potential against AML with WTH exhibiting more potent effects and should be explored further.

## 1. Introduction

Acute myeloid leukemia (AML) is the most common form of acute leukemia and a leading cause of cancer-related mortality [1]. AML incidence, currently at 4.3 new cases per 100,000 in the general population [2], is rapidly rising as the population ages because AML is more prevalent in the elderly [3]. Within a few weeks, AML can cause death if an effective treatment strategy is not adopted. Chemotherapy remains the most important tool for its treatment [4], although new treatments based on kinase and epigenetic inhibitors (namely, midostaurin, venetoclax, enasidenib and ivosinenib, among others) are now being adopted. These treatments, however, are restricted to the subpopulation of patients that present mutations on the *FLT3* gene for midostaurin or on *IDH1* or *IDH2* for ivosinenib or enasidenib, respectively [4,5,6,7]. Patients presenting these mutations represent <50% of total AML (~25% *FLT3*, ~12% *IDH2*, ~8% *IDH1* mutated). In most cases, these new therapies are not curative and, especially in patients >60 years, treatment related toxicity can lead to treatment-related mortality [8,9]. AML is, thus, a highly aggressive disease, and although drugs that target different pathways (e.g., *FLT3* tyrosine kinas, BCL-2 and *IDH1/2*) are now approved for the treatment of AML, with others being in different stages of development [9], overall survival rates are unfortunately still low, and show complications resulting from therapy and from the emergence of resistance [10]. This highlights the need for new therapeutic agents that can complement current targeted therapies to enable a sustained recovery of AML patients.

In the search for new therapies, the study of the medicinal properties of natural products is one of the fascinating fields in cancer therapeutics [11]. Several phytochemicals have been found to have anti-proliferative properties and number of studies have shown that these metabolites have potential to become novel therapeutic interventions for AML [12,13]. These compounds are usually effective in different phases of carcinogenesis by interacting with multiple molecular targets and cellular pathways [14]. *Datura innoxia* and *Withania coagulanse* are two important medicinal plants belonging to family Solanaceae. Extracts of both plants have been extensively investigated for treating different diseases and their putative anticancer properties are well-documented [15,16,17]. In an effort to identify the molecular causes of their reported antineoplastic properties, several withanolides were isolated from these plants and reported to have potent anti-proliferative activities in cancer cells [18], which may be due to their ability to induce apoptosis as well as the expression of tumor necrosis factor-α (TNF-α) activated nuclear factor-κB (NF-κB) [19]. However, the exact mechanism of action by which withanolides induce cancer cell killing is still unknown.

Protein phosphorylation is a common post transcriptional modification that is involved in the regulation of multiple biological processes. Phosphorylation events are modulated by protein kinases and their deregulation contributes to the onset and progression of essentially all types of cancer [20]. Therefore, targeting kinase mediated signaling pathways downstream of growth factor receptors (such as FLT3) is of high interest in the field of cancer therapeutics and these have been investigated to treat AML [21,22,23,24]. Receptor tyrosine kinases and downstream signaling pathways, such as phosphoinositide 3-kinase (PI3K) and mitogen-activated protein kinase (MAPK), are extensively reported for their role in oncogenic transformation and in regulation of cell cycle and apoptosis, all of which are well-known cancer hallmarks [25]. Indeed, the mechanism of action of several anti-cancer agents often involve induction of apoptosis by blocking pro-survival pathways regulated by protein and lipid kinases downstream of growth factor signaling.

In the present study, we evaluated the pharmacological potential of withametelin (WTH) and coagulansin A (CoA) in AML cell models and investigated their mode of action using proteomic and computational biology techniques. Our results indicate that WTH and coagulansin A withanolides have potent anti-leukemic activity by inhibiting several oncogenic pathways downstream growth factor receptors, including the PI3K/protein kinase B (PKB; also known as AKT) and MAPK pro-survival pathways, leading to apoptosis.

## 2. Materials and Methods

### 2.1. Cell Culture

AML cell lines, i.e., HL60, Kasumi-1 and P31/FUJ were cultured using RPMI medium (1640) supplemented with 10% FBS and 100 U·mL^−1^ penicillin/streptomycin. Cell confluency was maintained to 0.5–2.0 × 10^6^ cells·mL^−1^. The cells were kept in CO_2_ incubator at 37 °C under a humidified environment.

### 2.2. Viability Assays

Withanolides were isolated form *Datura innoxia* and *Withania coagulanse* through bioactivity guided isolation (unpublished data). Compounds were characterized through nuclear magnetic resonance spectroscopy (NMR) and identified as withametelin (WTH) and coagulansin A (CoA) isolated from *Datura innoxia* and *Withania coagulanse,* respectively. Compound purity was 99.8% as determined by high-performance liquid chromatography with a diode-array detector. To evaluate the antiproliferative potential of the isolated compounds, cells were seeded in 96 well plates (~10,000 cells/well). Cells were left for 3 h in incubator. Stock solutions of the compounds were prepared in dimethyl sulfoxide (DMSO) (Merck Life Science UK Limited, Gillingham Dorset, SP8 4XT, UK). First, single concentration, i.e., 40 µM was used for both the compounds, and then, different concentrations of WTH and CoA were used to calculate half maximal inhibitory concentration IC_50_ values. For WTH, the concentrations used were 500 nM, 1000 nM, 5 µM, 10 µM, 20 µM and 40 µM, while for CoA the concentrations used were 10 µM, 20 µM, 40 µM, 60 µM, 80 µM and 100 µM. Cells were treated for 48 and 72 h with the abovementioned concentrations. After attaining the required time of incubation, cells were stained with Guava ViaCount reagent (Millipore Corporation, Hayward, CA, USA) , https://www.merckmillipore.com) as indicated by the manufacturer and cell number, and viability was measured using a Guava EasyCyte Plus instrument (Millipore Corporation, Billerica, MA, USA). All experiments were performed in triplicate. IC_50_ values were calculated using the pharmacoGx package in R.

### 2.3. Apoptosis Assay

A Guava Nexin kit was used to assess apoptosis induced by withanolides according to the manufacturer’s protocol (Luminex Corporation 12212 Technology Blvd. Austin, TX, USA). AnnexinV-PE staining was performed as per the manufacturer’s protocol. Florescence was analyzed by cytosoft software. A minimum of 2000 events were counted.

### 2.4. Western Blot Analysis of Apoptosis Markers

To screen the markers of apoptosis, cells were treated with 15 µM of WTH and 60 µM of CoA and incubated for 1 h, 6 h and 24 h. Cells were thoroughly washed with cold PBS lysed using lysis buffer, composed of radioimmunoprecipitation assay buffer (RIPA) and inhibitors. Protein was quantified using a bovine serum albumin (BSA) assay. An aliquot of 60 mg protein was loaded to wells of sodium dodecyl sulphate-polyacrylamide gel electrophoresis (SDS-PAGE) and transferred to nitrocellulose membrane. Membrane was blocked with non-fat dry milk and immunodetected with primary antibodies from Cell signaling technology (CST) (Mabledon Place, London, UK) including poly (ADP-ribose) polymerase (PARP) (CST#9542), Bclxl (CST#2762), caspase-8 (CST#9746), caspase-9 (CST#9501), , caspase-3 (CST#9661) and anti-b-actin (CST#4967), and Bim (ab230531) from Abcam (Cambridge biomedical campus, Cambridge, UK), followed by secondary antibodies. The concentration of each antibody was used according to manufacturer’s protocol.

### 2.5. Proteome Analysis

#### 2.5.1. Cell Lysis and Protein Extraction

Almost 10 million cells of each cell line were seeded in T25 flasks and placed in incubator for 6 h and treated with 15 µM of WTH and 60 µM of CoA for 1 h and 6 h. Subsequently, the cells were harvested and washed three times with ice cold phosphate buffer saline (PBS), supplemented with 1 mM NaF and 1 mM Na_3_VO_4_. The experiment was performed in triplicate and pellets were preserved until all replicates were collected and further processed. For cell lysis, ice-cold urea lysis buffer was used, which was supplemented with 1 mM NaF, 1 mM Na_3_VO_4_, 1 mM β-glycerol and 1 mM Na_2_H_2_P_2_O_7_. Homogenization of lysates was carried out through sonication and insoluble material was pelleted out following centrifugation. The amount of protein in each sample was quantified using a BCA assay.

#### 2.5.2. Digestion

Samples were normalized to an equal protein concentration of 0.4 µg/µL. All samples were reduced using 10 mM dithiothreitol first and alkylated with 16.6 mM iodoacetamide. Each time, incubation at room temperature was carried out for 30 min with continuous shaking. Subsequently, urea concentration was reduced to 2 m by adding 20 mM 4-(2-hydroxyethyl)-1-piperazineethanesulfonic acid (HEPES). Tosyl-lysine chloromethyl ketone-trypsin beats were washed using HEPES and calculated amount was added into each sample. Samples were incubated overnight at 37 °C with agitation. Trypsin beats were pelleted out by centrifugation; the supernatant was collected for further processing.

#### 2.5.3. Desalting

The peptide solution obtained after digestion was transferred to ice. Desalting was carried out using OASIS 1cc HLB cartridges (Waters, UK). The cartridges were inserted into the manifold with vacuum up to ~5 in Hg and conditioned using 1 mL of 100% acetonitrile (ACN) followed by 1 mL desalting loading buffer (99%/1% H_2_O/ACN). This was equilibrated further with 0.5 mL of desalting loading buffer. Samples were loaded and eluted using the lowest possible flow rate. Subsequently, they were washed with 1 mL of desalting loading buffer. Samples were eluted using 0.5 mL 1M glycolic acid 50%/50% H_2_O/ACN + 5% trifluoroacetic acid (TFA) for phosphoproteomics and 0.5 mL 50%/50% H_2_O/ACN for proteomics. Proteomics samples were placed in speed-vac and dried overnight while phosphoproteomics samples were processed to next step.

#### 2.5.4. TiO_2_ Phosphopeptide Enrichment

For labeled free phosphoproteomics, samples were processed to phosphopeptide enrichment step. TiO_2_ beats were resuspended with 1% TFA. After vortexing, 25 µL of re-suspended TiO_2_ beads were added to each sample. Spin tips were placed in normal Eppendorf tubes and 200 µL 100% ACN was added to spin tips and centrifuged for 3 min at 1500× *g*, and the flow-through was discarded. The sample was added to the spin tips and centrifuged for 2 min at 1500× *g*, and the flow-through was discarded. After finishing the sample amount, 100 µL 1M glycolic acid in 80% ACN/5% TFA was added to each spin tip. Spin tips were centrifuged for 2 min at 1500× *g*. After discarding the flow-through, tips were washed with 100 µL 100 mM ammonium acetate. Next, three washings were carried out using 100 µL 90/10 H_2_O/ACN to spin tips. Using fresh Eppendorf tubes, samples were eluted with 5% NH_4_OH (10% ACN). Spin dips were discarded and samples were snap frozen in dry ice and subsequently placed in speed-vac and dried overnight.

#### 2.5.5. Nanoflow-Liquid Chromatography Tandem Mass Spectrometry (LC–MS/MS)

Each biological replicate was analyzed twice by nanoflow-liquid chromatography tandem mass spectrometry (LC-MS/MS). For phosphoproteomics each sample was reconstituted in 13 µL of reconstitution buffer. Samples were thoroughly mixed and incubated on ice for 20 min. Samples were then centrifuged and 12 µL was transferred to MS tubes. Samples were analyzed in a Q-Exactive Plus connected online to a Dionex 3000 nanflow liquid chromatograph equipped with a 0.075 × 500 mm column. Settings and conditions were as described before [26].

#### 2.5.6. MS Data Analysis

For analysis of MS data Mascot Daemon version 2.4 (Matrix Science Ltd, London, UK) with Mascot Distiller version 2.5.1.0 (Matrix Science Ltd, London, UK) was used. Mascot search engine processed files against the peptide sequence library in the SwissProt database restricted to Human entries (version 2010–03 containing 20,430 entries, UniProt, European Bioinformatics Institute (EMBL-EBI), Hinxton, Cambridge, UK; http://expasy.org/sprot/). Searches were restricted to human taxonomy, digested with trypsin enzyme. Carbamidomethyl (Cys) was used as a fixed modification, while oxidation, phospho S/T/Y and pyro-glu as variable modifications. Tolerance of 10 ppm and 25 ppm were the mass error tolerance thresholds for MS and MS/MS data, respectively. We used a script written in python to extract the results of Mascot and to perform quantitative analyses. The results were then transferred to excel files and further analysis was performed using the R statistical package. An unpaired *t*-test was used to infer statistical significance.

## 3. Results

### 3.1. Withametelin and Coagulansin A Inhibit the Proliferation of AML Models

To evaluate the effects of withanolide metabolites in human AML cells, we performed proliferation and cell viability assays in three representative AML cell lines treated with WTH or CoA. Both compounds inhibited the proliferation of AML cells in a dose-dependent manner (Figure 1). WTH was more potent than CoA in inhibiting both cell viability and proliferation; IC_50_ (mean ± SD) for WTH was 7.3 ± 3.7 µM (viability) and 2.0 ± 1.6 µM (cell numbers) (Figure 1A), and for CoA these values were 82 ± 13 µM and 16 ± 7.7 µM (Figure 1B), respectively. Thus, withanolide metabolites inhibited the proliferation and viability of AML cells at low micromolar concentrations, with WTH being >10 times more potent than CoA in these cell-based assays (Figure 1).

### 3.2. Withametelin and Coagulansin A Modulate Pathways Involved in Bioenergetic Metabolism, Signal Transduction and Regulation of Gene Expression 

To investigate the mode of action of WTH and CoA, we first performed a proteomic analysis of Kasumi-1 and P31-Fuj cells treated with 10 µM of WTH or 60 µM of CoA for 24 h. The experiment was performed with four replicates. We identified and quantified 35,476 unique peptides belonging to 5269 proteins with an FDR < 0.01 (Figure 2A). Metabolite treatment produced large changes in the proteome of these cells with WTH having a larger impact on the proteome than CoA (Figure 2B). Thus, WTH treatment reduced the expression of 323 and 455 proteins in P31-Fuj and Kasumi-1, respectively, whereas CoA reduced the expression of 96 and 126 proteins in these cells (Figure 2B). Both metabolites had a larger impact in reducing protein expression than in increasing expression.

Comparison of proteins decreased or increased by treatment in both cell lines showed that several proteins were consistently modulated by both metabolites (Figure 2C,D). These proteins included those involved in cell cycle control (such as CACL1 [27]), the regulation of gene expression (e.g., SETD2 and TTF2) and cell migration (e.g., PDLI2) [28].

Ontology and pathway enrichment analysis of the proteomics data (Figure 3) showed that the withanolides produced an overall increase of proteins in pathways associated with bioenergetic metabolism, such as the respiratory electron transport, respiratory electron chain and tricarboxylic acid cycle (Reactome and GO ontologies). Ontologies and pathways decreased by treatments were on the whole related to processes related to gene expression, ubiquitin biology and the regulation of translation (Figure 3).

To confirm the findings of the ontology and pathway analysis, we performed a targeted analysis of proteins in these ontologies (Figure 4). This analysis confirmed the upregulation of metabolic proteins (Figure 4A–E), including thioredoxin reductase 2 (TRXR2, involved in mitochondrial redox regulation), acetyl-coA aceteyltransferase (THIL, involved in mitochondrial beta-oxidation pathway), isocitrate dehydrogenase (IDHP, gene name *IDH2*), hexokinase 1 (HXK1) and 3-ketoacyl-CoA thiolase (THIM, also involved in beta-oxidation). We also observed a decrease in the expression of several signaling proteins (Figure 4F–J), including the protein kinases PAK2 (Figure 4F) and GSK3A (Figure 4I), which were decreased by both treatments in both cell lines. This targeted analysis also confirmed an effect of the compounds in the ubiquitin-metabolizing proteins (Figure 4P–S). Overall, these data indicate that WTH and CoA induce changes in the expression of proteins involved in regulating key cellular processes including metabolism, signal transduction, protein degradation and gene expression.

### 3.3. Phosphoproteomic Analysis Reveals That Withametelin and Coagulansin A Have an Impact on the Activity of Pro-Survival Kinase Signaling

To elucidate signaling pathways that may be involved in the withanolides’ mode of action, we performed a phosphoproteomic analysis of cells treated with 15 µM of WTH or 60 µM of CoA for 1 h or 6 h in four replicates (Figure 5A). This experiment quantified 17,482 phosphosites to 4284 proteins (Figure 5B). The impact of WTH on the phosphoproteome was greater than that of CoA (Figure 5C), consistent with their effect on the proteome.

Enrichment analysis of the phosphoproteomics data against kinase substrates (Figure 5D), kinase-kinase relationships (Figure 5E) and canonical pathways (Figure 5F) showed that the withanolides decreased the activity of kinases such as PI3K (PIK3CA), AKT1, mTOR, ERK1/2 (MAPK1/3) and serine/threonine-protein kinase A-Raf (ARAF) (Figure 5D,E). Our data also showed an increase in the phosphorylation of proteins in the caspase cascade in apoptosis, P53 and DNA-PK pathways (Figure 5F).

To confirm the results of kinase substrate and pathway enrichment analysis, we analyzed the expression of individual phosphorylation sites known to be markers of pathway activity. Consistent with the enrichment data, we found that the withanolides decreased phosphorylation of AKT1, AKT2, MTOR, ERK (gene names MAPK1 and MAPK3) (Figure 6A,B) and increased phosphorylation of kinases involved in DNA repair such as DNA-PK (gene name PRKDC) and ATM, and the DNA damage marker H2AFY at Ser170 (Figure 6C). The phosphorylation of several other signaling proteins such as STAT3, STAT5, PAK and PKC isoforms, and GSK3B were also modulated by CoA and WTH treatments (Figure 6D). Together, these results indicate that withanolide metabolites have pleiotropic effects in decreasing multiple pro-survival signaling cascades and in the activation of pro-apoptotic pathways.

### 3.4. Withametelin and Cagulansin A Induce Apoptosis in AML Cells

To functionally confirm that the withanolides induce apoptosis in AML cells we measured apoptotic markers as a function of treatment with these metabolites (Figure 7). We observed a cleavage of PARP and Caspases 3, 8 and 9 and an increase of Bim after 6 h of treatment, thus confirming that the inhibition of pro-survival pathways by the withanolides results in an induction of apoptosis in AML cells.

## 4. Discussion

Here, we investigated the mode of action of two potentially valuable natural compounds, i.e., WTH and CoA, and propose that they may provide a new route for the treatment of AML. The major findings of the current study are: (i) WTH and CoA possess potent anti-proliferative and pro-apoptotic activities against AML cell models; (ii) WTH has more potent cytotoxic activity than CoA; (iii) both withanolides modulate the expression of proteins involved in regulating key cellular processes including metabolism, signaling, cell cycle, protein degradation and gene expression; and (iv) both withanolides block PI3K and MAPK oncogenic pathways and increase the phosphorylation of proteins in the caspase cascade leading to apoptosis.

Previously, we reported that *Datura innoxia* is a rich source of phytochemicals with cytotoxicity against human leukemia cell lines [29]. We also found that withanolides from *Withania coagulanse* exhibited the potential to inhibit NF-κB [30]. We isolated WTH from *Datura innoxia* and CoA from *Withania coagulanse* [30]. The results from a number of studies show that withanolides have the potential to induce apoptosis in both myeloid and lymphoid cells along with primary cells derived from leukemia patients [31,32]. These studies motivated us to evaluate the isolated withanolides for their anticancer cytotoxic activities against AML. We found that both the compounds not only inhibited the proliferation of AML cell lines but also induced cell death. With relevance to WTH, more promising effects were observed in P31-Fuj and Kasmi-1 cells as compared to HL60 might be due to differences in their genetic backgrounds. AML cells have highly heterogeneous genetic and epigenetic backgrounds and, consequently, their responses to therapy are also highly variable. Further studies are needed to identify the patient subpopulations more likely to respond to therapies based on withanolide treatment and whether such treatments have sufficiently wide therapeutic windows.

In order to rationalize withanolides’ cytotoxic activities, different mechanisms of action have been proposed. However, it remains unclear which of them apply across different metabolites and are more relevant in different models. In this study, we employed high-content quantitative proteomics to determine the mechanisms by which the withanolides WTH and CoA exert their anti-proliferative effects. We found significant changes in the proteome of treated cells as compared to untreated controls, with 33 proteins showing decreased expression in CoA-treated cells as compared to 53 affected in WTH treated samples. As described in the results section, our proteomics analysis revealed that withanolides mainly modulate the expression of proteins, which involved multiple key pathways relevant to cancer growth, metabolism, survival and proliferation. We found large differences in how WTH and CoA affect protein expression, thus showing that these two metabolites modulate cell biology through different mechanisms. An interesting finding is that WTH and CoA have different effects in cellular metabolism. Since tumor cells cannot compensate for the inhibition of the energy shunt that is required for tumor cell growth and progression, starving cells for their energy requirement will ultimately lead to apoptosis. We found that the major metabolic proteins inhibited by WTH or CoA were TRXR2, THIL, IDHP, HXK1 and THIM, which are involved in redox-regulated cell signaling and in the bioenergetic metabolism of carbohydrates and lipids.

Kinase signaling pathways, which are a rich source of targets for anticancer drug development, were also inhibited by withanolide treatment. Interestingly, we observed a significant reduction in the expression of PAK2, a serine/threonine protein kinase that plays a critical role in variety of signaling pathways with roles in cytoskeleton regulation, cell cycle, motility, apoptosis and proliferation [33]. Other major proteins related to kinase signaling and showing significant differential expression in withanolide-treated cells include IBP2, GSK3A, GSK3B and CKS2. These proteins have diverse functions in mediating growth factor signaling, development and cell cycle regulation [34]. Interestingly, expression of IBP2 and GSK3B decreased in CoA treated cells and increased after WTH treatment (Figure 4). The withanolides also affected proteins involved in transcription regulation and the ubiquitin system; these include TMA7, IF4H, IF4B, IF1AX and IF4E and UBP2L, UB2L3, UFL1 and UBE4A, respectively. Thus, the withanolides modulated the expression of proteins in multiple pathways that are considered to be cancer hallmarks [35], highlighting the potential of these metabolites for the treatment of cancer.

To complement the proteomics data, and to investigate the effects of withanolides in signaling in greater detail, we performed a phosphoprotemics analysis of WTH and CoA treated AML cells. KSEA was used for estimation of kinase activity as a function of treatment [36]. We found that the withanolide metabolites inhibited the phosphorylation and activity of major proteins involved in kinase signaling which include PI3K, AKT1, mTOR, ERK1/2 and ARAF. In previous studies, withanolide D was proposed to modulate the phosphorylation of JNK and p38MAPK and consequently induce apoptosis in myeloid and lymphoid cells [31]. PI3K/Akt pathway is one of the major pathways targeted by anticancer drugs, and studies have shown that inhibition of the PI3K/AKT pathway leads to decreased cell proliferation in AML [37,38]. The decrease in activity of mTOR and PIK3CA was highly significant for both the compounds at 1 h and 6 h treatment. We observed after 1 h of treatment with CoA that there was increased phosphorylation of proteins involved in MAPK signaling, which was decreased at 6 h. This might be due to the activation of feedback loops at 1 h. Interestingly, we found that both WTH and CoA inhibited the activity of the mitotic kinase TTK (also known as Monopolar spindle 1 (MPS1)). Recently, we found that TTK inhibition induced more cell death than kinase inhibitors against other targets showing that kinase TTK is important in AML [26]. Together, these findings further underscore the potential of WTH and CoA for the treatment of AML.

Apoptosis can be triggered through multiple signaling pathways, but the ultimate event of physiological or chemotherapy-induced apoptosis involves the permeabilization of mitochondrial membrane and the consequent release of caspases. We found that this caspase cascade was enhanced in withanolide treated cells thus confirming that these metabolites induce apoptosis in AML. This was established by immunoblotting showing the cleavage of caspase 9 at 6 h and 24 h of treatment, while caspase 8 expression was observed at 6 h. Caspase was activated in both the compounds in both the cells lines. We also observed the decreased expression of Bcl-xl, an anti-apoptotic protein, and increased expression of bim, a pro-apoptotic protein. Senthil et al. reported a similar mechanism for the action of withanolide isolated from *Withania somnifera* in AML [39]. Our results are also in line with a study of Mondal et al., who reported that withanolide D induces apoptosis in leukemic cells and primary cells derived from leukemia patients through modulation of stress kinases [31]. Similarly, purified withaferin A, isolated from the Aswagandha plant, is reported to reduce the levels of Bcl-xl and inhibition of the activation of Akt in leukemia after 24 h of treatment [40]. In our study the induction of late apoptotic events was also reflected by detection of cleaved PARP and this cleavage is the useful hallmark of apoptosis (Figure 7).

In conclusion, this study shows that the withanolide metabolites WTH and CoA inhibit the proliferation of AML by targeting oncogenic kinase signaling pathways that regulate cell cycle progression and survival, thus resulting in an induction of apoptosis. Our results provide new insights into the molecular mechanism by which withanolides exert their potential anticancer activities.

## Figures and Tables

**Figure 1 biomedicines-08-00333-f001:**
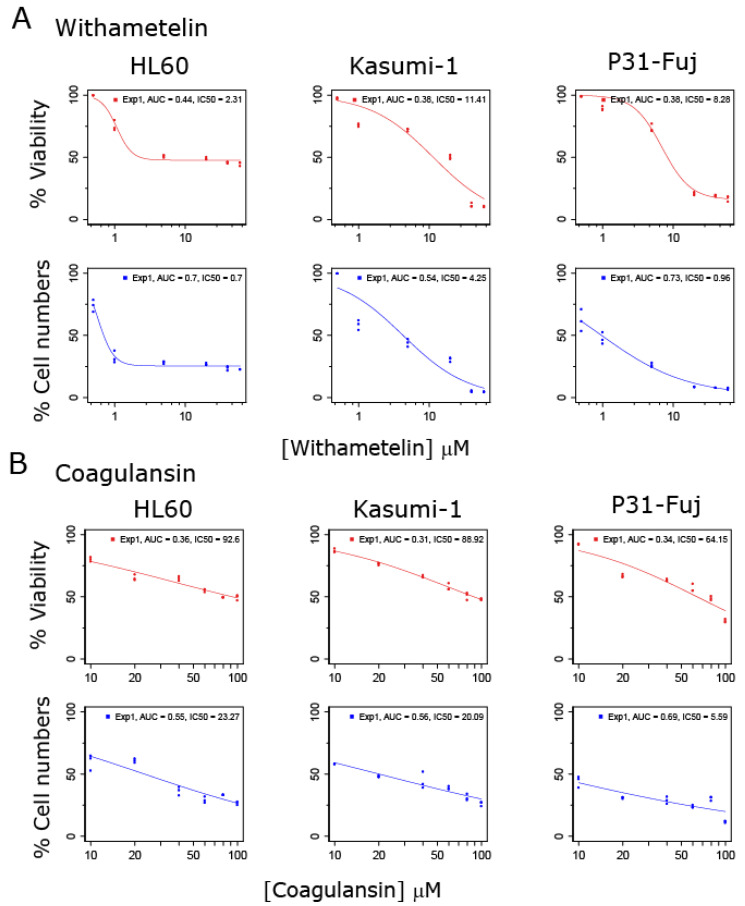
Impact of withanolides in reducing cell viability and proliferation of acute myeloid leukemia (AML) cell models. Percentage of viable cells and number of cells relative to control was determined as a function of treatment with withametelin (WTH) (**A**) or coagulansin A (CoA) (**B**) in the named cell lines. IC_50_ was determined using the pharmacoGx package in R.

**Figure 2 biomedicines-08-00333-f002:**
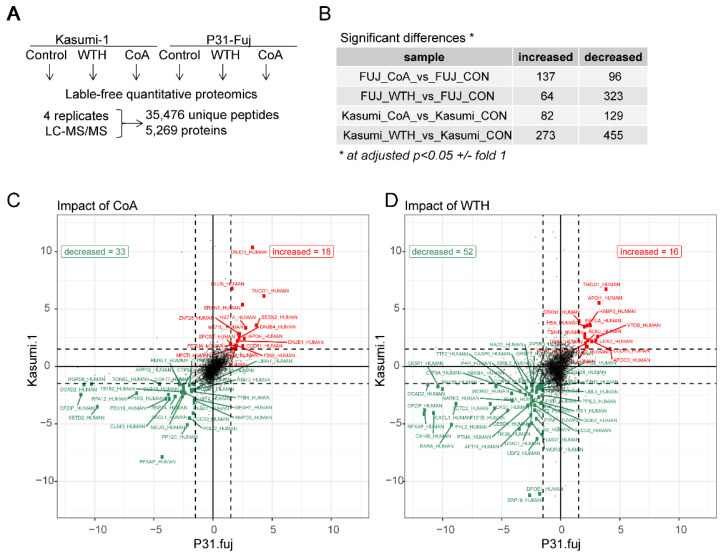
Proteomic analysis of AML cells treated with withanolide metabolites. (**A**) Experimental design and summary of identified proteins. (**B**) Number of proteins increased or decreased as a result of treating the named cell lines with CoA or WTH. (**C**,**D**) Comparison of protein increased or decreased by CoA (**C**) or WTH (**D**) across the tested cell lines. Proteins increased as a result of treatment are shown as red data points, while decreased proteins are shown are green data points.

**Figure 3 biomedicines-08-00333-f003:**
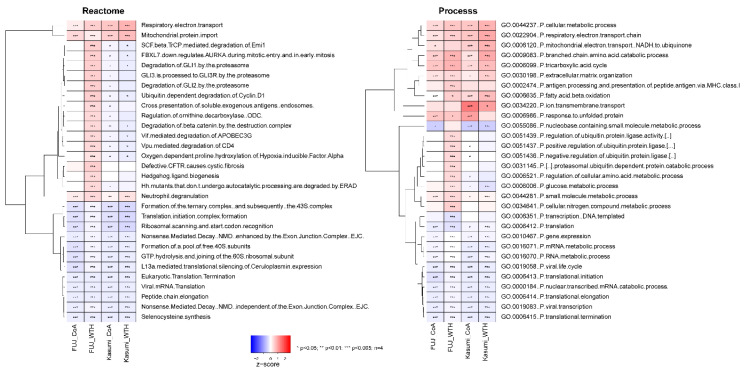
Pathway and ontology analysis of proteins modulated by withanolides. Z-scores and *p*-values were calculated for the named cell lines and treatment using the parametric analysis of gene expression (PAGE) and Kolmogorov-Smirnoff test, respectively (see Methods for details).

**Figure 4 biomedicines-08-00333-f004:**
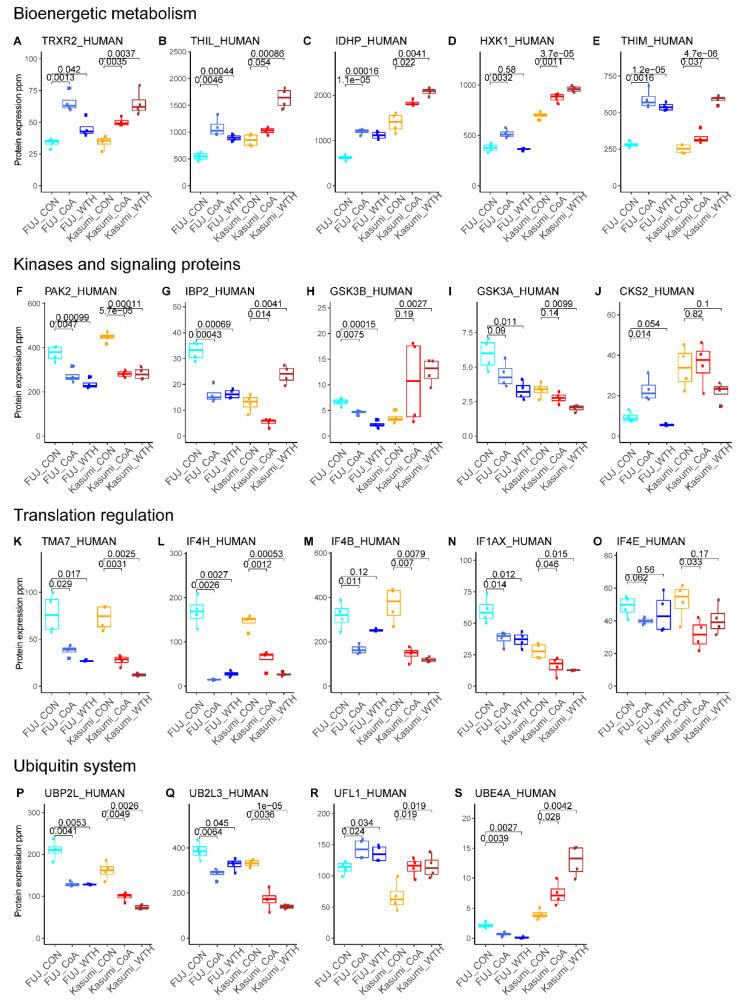
Withanolides produce changes in expression of proteins with key roles in cell biology. Shown are a selection of proteins with roles in bioenergetic metabolism (**A–E**), kinase signaling (**F–J**), translation (**K–O**) and the ubiquitin system (**P–S**). Numbers linking conditions denote *p*-values obtained by *t*-test (*n* = 4). Data points showing treatments to P31-Fuj are shown in shades of blue; treatments to Kasumi-1 cell lines are known in shades of red/brown.

**Figure 5 biomedicines-08-00333-f005:**
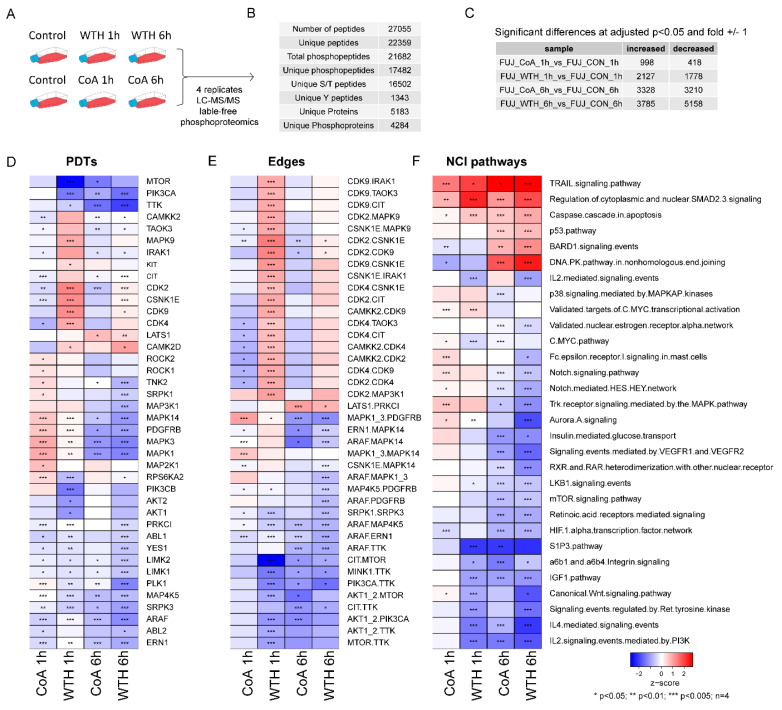
Phosphoproteomic analysis of withanolide-treated cells. (**A**) Experimental design. (**B**) Summary of quantitative data. (**C**) Number of phosphopeptides changes induced by WTH and CoA at the named time points. (**D–F**) CoA and WTH induced changes in kinase activities (**D**), kinase axes (**E**) and pathways (**F**). Z-scores and *p*-values were calculated for the named cell lines and treatment using the PAGE and Kolmogorov-Smirnoff test, respectively (see Methods for details).

**Figure 6 biomedicines-08-00333-f006:**
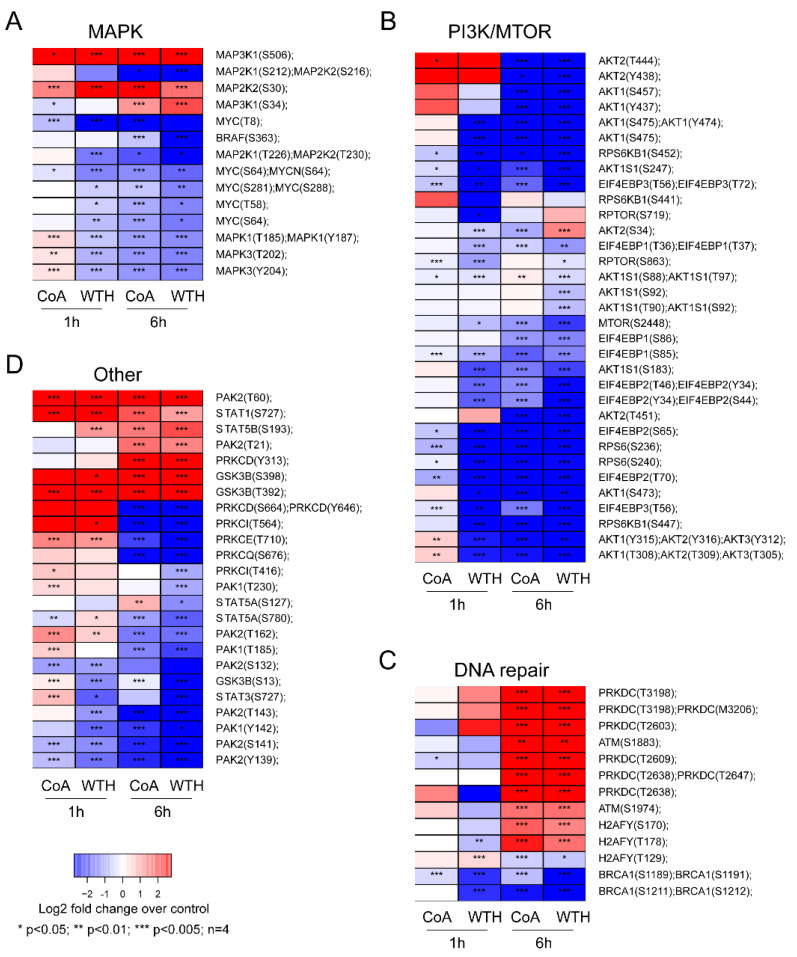
The withanolides induce pleiotropic effects in the phosphorylation of pro-survival signaling proteins. Shown are changes in phosphorylation sites in mitogen-activated protein kinase (MAPK) (**A**), phosphatidylinositol 3-kinase/ mammalian target of rapamycin (PI3K/MTOR) (**B**), DNA repair (**C**) and other signaling pathways (**D**) induced by withanolide treatment at the indicated time points. *p*-values were calculated by *t*-test (*n* = 4).

**Figure 7 biomedicines-08-00333-f007:**
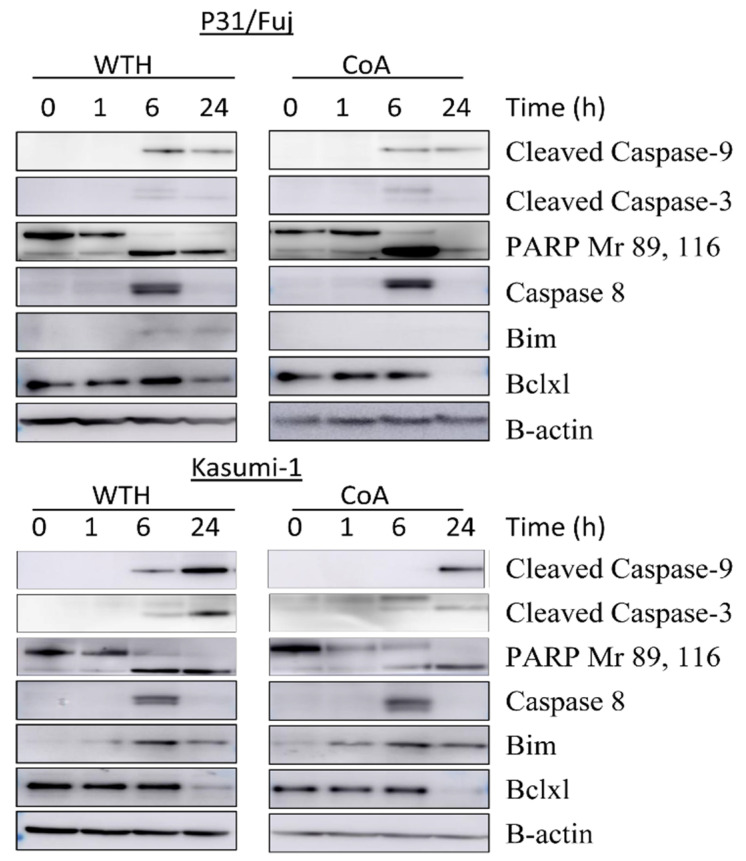
The withanolides induce apoptotic markers. Western blot images of the named markers are shown. See methods for details of antibodies used.

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
