# Peer review of "Withanolide Metabolites Inhibit PI3K/AKT and MAPK Pro-Survival Pathways and Induce Apoptosis in Acute Myeloid Leukemia Cells"

_biomedicines, 2020, doi:10.3390/biomedicines8090333_

Round 1

Reviewer 1 Report

This is an interesting report in which the authors developed the novel therapeutic potential agents against AML. The authors in this report thoroughly describe that Wthanolides metabolite include withametelin (WTH) and Coagulansin A (CoA) inhibit the proliferation of AML by targeting oncogenic kinase signaling pathways, resulting in an induction of apoptosis.

Although the manuscript with clearly figure and table is well written in each section, several points should be amended or clarified for the benefit of the readers.

Minor considerations;

  1. The authors should show any influences for human normal hematopoietic stem cells ( ex; CB CD34+) and peripheral blood of WTH and CoA.
  2. Could you check the effects of them to doxorubicin and/or cytarabine resistant AML cell line? It might be very helpful to understand the significance of this paper.

Author Response

This is an interesting report in which the authors developed the novel therapeutic potential agents against AML. The authors in this report thoroughly describe that Wthanolides metabolite include withametelin (WTH) and Coagulansin A (CoA) inhibit the proliferation of AML by targeting oncogenic kinase signaling pathways, resulting in an induction of apoptosis.

Although the manuscript with clearly figure and table is well written in each section, several points should be amended or clarified for the benefit of the readers.

> We thank the Reviewer for the positive and constructive review. 

Minor considerations;

Point 1: The authors should show any influences for human normal hematopoietic stem cells ( ex; CB CD34+) and peripheral blood of WTH and CoA.

Response1: This is an important point and we thank the reviewer for making this suggestion. We are carrying out these experiments now. Our preliminary data show that Withametelin cytotoxicity against isolated human normal lymphocytes is 57.23±1.22% (as percent inhibition) with an IC50 of 14.65±1.31 µg/ml.  It was much greater than vincristine (positive control) which showed percent inhibition of 67.73±1.70% with IC50 of 6.86±0.49 µg/ml. 1% DMSO did not impact cell viability. Given the short time lines for replying to this review, these data will be submitted in a different paper along with cytotoxicity evaluation against several other cancer cell lines. Furthermore, Coagulansin A was checked in a previous manuscript of a coauthor against articular chondrocytes which are normal cells and it did not show any toxicity [1].

[1] Phull, Abdul Rehman, Mubshir Hassan, Qamar Abbas, Hussain Raza, Ihsan ul Haq, Sung Yum Seo, and Song Ja Kim. "In Vitro, In Silico Elucidation of Antiurease Activity, Kinetic Mechanism and COX‐2 Inhibitory Efficacy of Coagulansin A of Withania coagulans." Chemistry & Biodiversity 15, no. 1 (2018): e1700427.

Point 2: Could you check the effects of them to doxorubicin and/or cytarabine resistant AML cell line? It might be very helpful to understand the significance of this paper.

Response 2: This is another interesting suggestion. We would like to note that the main objective of the study was to investigate the mode of action of withametelin and Coagulansin A in AML, which has not been studied previously. We chose one cell line (P31-Fuj) which is relatively resistant to cytarabine. We agree that further work is required to understand the AML subgroup that may be more sensitive to therapies based on withametelin and Coagulansin A (we make this point in the Discussion section) and we plan to evaluate these compounds in different resistant AML cell models. We however consider that these experiments are outside the scope of the present investigation, which, as mentioned above, was mainly concerned with the investigation of the mode of action of these metabolites in AML. In this respect, we provide the community with extensive novel proteomics and phosphoproteomics data showing that withametelin and Coagulansin A inhibit AML cell viability by inhibiting key signalling pathways that regulate apoptosis.

Reviewer 2 Report

This is an interesting paper reporting anticancer activities of  Withanolides. The work is well done using proteomic and bioinformatic techniques, as well as a number of experimental cellular biological methods. The manuscript is written logically, and will be interesting to readers of the journal. The authors should concern the following points:

  1. What is the purity of the compounds withametelin (WTH) and coagulansin A (CoA) isolated from Datura innoxia and Withania coagulans?
  2. The methods should be described in detail. Antibodies for Western blot analysis of apoptosis markers – where manufactured?
  3. Page 10, line 8. “we observed a cleavage of PARP and Caspases 3, 8 and 9…” No cleaved Caspase 8 in Figure 7; and in Kasumi-1 cells the Cleaved Caspase-9 was detected only at 0 h time - ?
  4. The amount of the experimental data on pro-apoptotic activity of the compounds WTH and CoA is a little bit weak, it would be good to compliment Western blot data with flow cytometry data as well, using Annexin-V and Propidium Iodide double-staining or assessment of DNA fragmentation by PI staining of hypo-diploid nuclei.

Page 3:

Line 35, line 41, mM - ?

Line 43, M -? mM -?
